# Pre-Anticipatory Anxiety and Autonomic Nervous System Response to Two Unique Fitness Competition Workouts

**DOI:** 10.3390/sports7090199

**Published:** 2019-08-27

**Authors:** Gerald T. Mangine, Brian M. Kliszczewicz, Joseph B. Boone, Cassie M. Williamson-Reisdorph, Emily E. Bechke

**Affiliations:** Department of Exercise Science and Sport Management, Kennesaw State University, Kennesaw, GA 30144, USA

**Keywords:** heart rate variability, high-intensity functional training, recreational athletes, catecholamines, CrossFit

## Abstract

To evaluate the feasibility of on-site collection of subjective anxiety, autonomic nervous system activity, and salivary catecholamines surrounding high-intensity functional training (HIFT) competition, ten experienced HIFT competitors completed baseline assessments of anxiety and heart rate variability (HRV). Then, in two consecutive weeks (Workout 1 and 2) within the competition, HRV was recorded and examined in 5-min segments prior to exercise (PRE) and across a 30-min period after competitors completed their choice of the prescribed or scaled each workout. Subjective anxiety ratings and saliva samples were collected at PRE and immediately-(IP), 30-min (30P), and 60-min post-exercise (60P). Saliva samples were analyzed for concentrations of epinephrine and norepinephrine. Generalized linear mixed models with repeated measures revealed significant (*p* < 0.05) differences between workouts for all measures. Compared to Workout 1, anxiety (~50%), epinephrine (173–340%), norepinephrine (29–234%) were greater in Workout 2 and various HRV-derived indices were more depressed. Additionally, some HRV-derived indices appeared to be modulated (*p* < 0.05) by competitive level and sex at PRE and throughout the 30-min recovery period. These data suggest that autonomic activity may differ between the competitive and laboratory settings, and that the response may be further modulated by the workout’s design, the athlete’s sex, and competitive level.

## 1. Introduction

In sports and training, success (or failure) can be heavily influenced by the manner and rate of recovery that occurs between competitive events or workouts [1]. The most popular competition, featuring high-intensity functional training (HIFT), is an annual, international, multi-stage event used to identify ‘the fittest on earth’ held by CrossFit®. It begins with an online fitness competition (i.e., the Open) that consists of several novel workouts that are released individually over a 5-week period. A unique feature of the Open is that it permits unlimited attempts per workout within a four-day window. Multiple maximal efforts might lead to prolonged perturbations to homeostasis [1,2]. Considering that competitors are likely to continue training during this time, delayed recovery may negatively impact normal training habits, competition performance in later weeks, or both. More importantly, the combination of training and multiple, maximal efforts over a prolonged period may be a recipe for overtraining [3,4,5], particularly in individuals who are not accustomed to the demands of HIFT competition (i.e., recreational athletes).

The ability to recover from HIFT workouts and competitive events is not well understood. The results of a few studies examining markers of oxidative stress [6] and inflammation [7,8], following various HIFT protocols, have been equivocal. Moreover, examining acute changes in these biomarkers may not be relevant because of their potential contributing roles in the muscular adaptation process [9] and because their response may diminish over time (i.e., repeated bout effect) [10]. In one of these studies [7], the researchers used a more practical measure (i.e., power) to complement the assessed biomarkers and found that mean and peak power during the back squat was either maintained or improved within 24 h following two consecutive daily workouts, respectively. However, this assessment may have lacked relevance because neither of the workouts included this movement and it may not adequately represent systemic recovery. Another way to examine general readiness and systemic recovery is to measure autonomic nervous system activity. This can be accomplished non-invasively by recording heart rate variability (HRV) and quantifying circulating catecholamines (epinephrine, norepinephrine). HRV measures the timing between consecutive inter-beat intervals via an electrocardiogram or beat-to-beat calculating device. Various HRV-derived indices (e.g., high-frequency domain (HF) of power spectral density, root mean squares of successive normal-to-normal differences (RMSSD)) are sensitive to physical and emotional stress (i.e., anxiety) [11,12,13] and reflect parasympathetic activity. Their depression indicates normal autonomic nervous system modulation in response to systemic demands brought on by physical stress (e.g., exercise), whereas their rate of rebound suggests homeostatic gain and systemic readiness. In conjunction to parasympathetic modulation, sympathetic neurohormones (i.e., epinephrine and norepinephrine) largely govern cardiovascular activity and are known to elevate surrounding physical activity [14]. Monitoring HRV and catecholamine concentrations together may provide a holistic view of the autonomic nervous system activity in response to HIFT and subsequent recovery. 

Circulating catecholamines and HRV-derived indices have been monitored surrounding benchmark workouts “Cindy” (pull-up, push-up, bodyweight squat circuit) [6] and “Grace” (30 clean and jerks) [15], as well as a 15-min HIFT circuit (row, kettlebell swing, thruster) [15]. Although these studies suggest that autonomic nervous system activity was recovered within 1–2 h of exercise [6,15], these findings lack practical context. The participants engaged in familiar HIFT protocols within a controlled, laboratory setting on a convenient day and time. In contrast, HIFT competition workouts and workouts of the Open are predominantly novel and all attempts for a specific workout must occur within a pre-set time frame. Although these factors (i.e., novelty, time limitation) and the competitive setting are known to affect anxiety and autonomic nervous system function [14,16,17], no study has confirmed this in relation to HIFT. Therefore, the purpose of this exploratory study was to evaluate anxiety and autonomic nervous system activity surrounding two consecutive HIFT workouts occurring during the Open in a natural competitive environment. This observational study will provide insight into the feasibility of evaluating autonomic nervous system recovery in non-laboratory settings.

## 2. Materials and Methods

### 2.1. Participants

Male (*n* = 5, 34.4 ± 3.8 years, 176 ± 5 cm, 80.3 ± 9.7 kg, 14.4 ± 3.1% fat) and female (*n* = 5, 35.5 ± 7.0 years, 159 ± 7 cm, 76.9 ± 21.4 kg, 29.4 ± 10.9% fat) recreationally-trained HIFT practitioners who were enrolled to compete in the Open were recruited for this study. Following an explanation of all procedures, risks and benefits, each participant provided their written informed consent to participate. The study was conducted in accordance with the Declaration of Helsinki, and the protocol was approved by the Kennesaw State University Institutional Review Board (#16-242). All participants were free of any physical limitations (determined by medical history questionnaire and PAR-Q) and had been regularly participating (at the time of recruitment) in HIFT for a minimum of 2 years. 

### 2.2. Study Design

Within two weeks of the Open, prospective participants arrived at the Human Performance Laboratory (HPL) in the morning, 3–4 h post-prandial and having avoided physical activity (24 h) to complete enrollment and to complete anthropometric and baseline assessments of anxiety and resting autonomic nervous system function. To more accurately evaluate the response of baseline measures as they occur during the Open, the remainder of the investigation was completed at the same local gym where each participant was a member. In two consecutive weeks of the competition, participants arrived approximately one hour before they were scheduled to complete the specific week’s competition workout. Both workouts were completed at mid-day (12:00–3:00 pm EST), approximately 18–40 h after competition officials released their details. Upon arrival, participants were fitted with a Polar Team^2^ heart rate (HR) monitor (Polar USA, Lake Success, NY, USA) and rested for 10-min before providing pre-exercise (PRE) saliva samples and subjective anxiety rating. Participants continued to wear the HR monitor throughout their self-selected warm-up, the workout, and for 30 min post-exercise. Saliva samples and subjective assessments of anxiety were collected again immediately (IP), 30-min (30P), and 60-min post-exercise. All HR data was treated to examine HRV at rest and in response to each workout, while saliva samples were analyzed for concentrations of epinephrine and norepinephrine. 

### 2.3. Anthropometric Assessments

Height (± 0.1 cm) and body mass (± 0.1 kg) were determined using a stadiometer (WB-3000, TANITA Corporation, Tokyo, Japan) with the participants standing barefoot, with feet together, in their normal daily attire. Body fat percentage was determined using whole body-dual energy x-ray absorptiometry (DXA) scans (iDXA; Lunar Corporation, Madison, WI, USA) in “standard” mode and using the company’s recommended procedures and supplied algorithms. Quality assurance was assessed by daily calibrations performed prior to all scans using a calibration block provided by the manufacturer.

### 2.4. Fitness Competition

The Open is a 5-week competition where officials release workouts individually on each Thursday (5:00 pm pacific time) and competitors may complete multiple attempts before submitting their best attempt online by the following Monday (5:00 pm pacific time). Workouts generally consist of two or more exercises performed for a prescribed (Rx) number of repetitions and sets using standardized technique and/or resistance. A “Scaled” option, which consists of modified technical requirements, volume, and/or reduced resistance, is also made available. For the present study, participants were asked to select the workout option (Rx or Scaled) that would allow them to “perform their best”, and only their first attempt was considered to minimize the potential influence of fatigue or workout familiarity on the autonomic response. To examine autonomic nervous system activity over time, two consecutive workouts (i.e., two consecutive weeks) were selected for analysis. Although data was collected on all five workouts, fair comparisons would not have been possible due to various circumstances (i.e., Workout 1 = Incomplete data set; Workout 2 = Workout structure; and Workout 5 = Participants completed at night) beyond the control of the investigators. In contrast, data was collected in all participants on the third (Workout 1) and fourth weeks (Workout 2) and the duration of these workouts closely resembled those of previous HIFT investigations on autonomic nervous system activity [6,15]. Scoring for both workouts was based on the competitors’ ability to complete ‘as many repetitions as possible’ (AMRAP) within a set time frame. All workouts were completed at the same gym in front of an official judge. The descriptions, standards, and scoring criteria for each workout are available on the competition website [18] and are summarized in Table 1.

### 2.5. Anxiety

Anxiety towards the competition and workout were assessed via a single-item Likert scale that consisted of five evenly spaced values, each anchored to a level of anxiety (1 = not at all anxious, 2 = a little anxious, 3 = moderately anxious, 4 = very anxious, or 5 = extremely anxious). The participant was asked to circle the number that best described how anxious they felt at that moment. Values were scored in arbitrary units (au). The validity and reliability of a single-item Likert scale to assess current anxiety has previously been established [19].

### 2.6. Saliva Sample Collection and Analysis

Participants were asked to refrain from consuming any food, drinking hot fluids, or brushing their teeth for two h prior to their arrival. Upon arrival, participants were asked to remain seated for 15 min before providing their resting sample (i.e., PRE-sample). All participants then completed a self-selected warm-up for approximately 10–15 min before attempting the workout. Participants provided their IP sample within one min of workout completion and were then asked to remain in a relaxed position (e.g., seated or standing) at the training facility for 60 min. The participants could drink water ad libitum while waiting but were asked to refrain from drinking water within 10 min of any post-exercise collection time point.

Approximately 2 mL of saliva were collected into a cryovial (Salimetrics LLC, State College, PA, USA) in duplicate at each time point using the passive drool method and stored at −80°C until assay. Concentrations of epinephrine and norepinephrine were assessed via commercially-available, enzyme-linked immunosorbent assays (ELISA) and a spectrophotometer (SpectraMax M3, Molecular Devices, Sunnyvale, CA, USA). To eliminate inter-assay variance, all samples were thawed once and analyzed in duplicate in the same assay run by a single technician, with an average coefficient of variation (CV) of 7.6% for epinephrine and 7.8% for norepinephrine. 

### 2.7. Heart Rate Variability

Heart rate was recorded via the Polar Team^2^ system (Version 1.4.5, Polar USA, Lake Success, NY, USA) during the baseline visit and surrounding each workout. The system consists of a Polar monitor strap worn around the chest against the skin that was affixed with a transmitter that connected through Bluetooth to a base station. During the baseline visit and prior to each workout session, participants were placed in a quiet, dimly lit room and instructed to remain as still as possible while in a seated position for 10 min. Following each workout, the system continued to collect heart rate data for 30 min. 

The obtained recordings were divided into 5-min segments for analysis: baseline and PRE-recordings measured the last 5-min of the 10-min recordings; the 30-min post recording was divided into five, 5-min segments at 5–10, 10–15, 15–20, 20–25, and 25–30 min. The presence of artifact noise was filtered through a piecewise cubic spline interpolation method using a “low artifact correction” with a sensitivity set to identify any R-R abnormality ±0.35 sec compared to the local average through a function available in the Kubios software (Version 2.2, Kubios, Joensuu, Finland) [20,21]. To avoid analysis distortion, segments containing three or more irregular R-R intervals (e.g., artifact or ectopic beats) [22] were excluded.

HR was calculated as a percentage of age-predicted maximum (%HR_MAX_; 220-age) [23] and averaged over the last 5 min of PRE, the duration of exercise, and across 5-min intervals from 5–30 min post-exercise. HRV measures were obtained from the same 5-min segments (except during exercise) and quantified as the natural log (ln) of RMSSD, the frequency domain index of high-(lnHF; 0.15–0.40 Hz) and low-frequency (lnLF; 0.04–0.15 Hz) filter of the power spectral density, and the LF:HF ratio. Acquired R-R interval recordings were transformed into time and frequency domain components using specialized online HRV software (Version 2.2, Kubios, Joensuu, Finland). To assess RMSSD, R-R intervals were converted into a tachogram, which plots the successive R-R intervals (*y*-axis) against the number of beats within the total number of beats in the recording (*x*-axis). Five-min recordings were sampled from the tachogram to analyze RMSSD. HF and LF were analyzed through power spectral analysis through the application of a fast Fourier transformation of the R-R interval recording with a window width of 256 s and overlap of 50%. RMSSD and HF are widely accepted markers of parasympathetic activity and are commonly used to assess vagal activity following exercise, whereas LF is believed to be an indication of sympathetic and vagal activity [11,12,13]. 

### 2.8. Statistical Analyses

Changes in anxiety, HRV-derived indices, and epinephrine/norepinephrine concentrations were separately examined across time using a generalized linear mixed model with maximum likelihood estimation and an autoregressive-heterogenous repeated covariance to account for the dependent relationships existing between time points. Due to differences in workout prescription, sex and competitive level (i.e., Rx or Scaled) were added as factors into the model. Following any significant F-ratio, specific differences were further assessed by applying adjustments to confidence intervals using the least significant difference procedure. All differences between time points were further analyzed by effect sizes calculated according to Cohen’s *dz* [24]. As previously suggested for recreationally-trained individuals [25], interpretations of effect size were evaluated at the following levels: trivial (<0.35), small (0.35–0.80), moderate (0.80–1.50), and large (>1.50). All data are reported as mean ± standard error. SPSS statistical software (version 25, SPSS, Chicago, IL, USA) was used for all analyses with statistical significance set at *p* < 0.05. 

## 3. Results

Performances on Workout 1 and Workout 2 for all competitors are presented in Table 1. Significant sex x time (F = 2.06, *p* = 0.029) and competitive level x time (F = 3.50, *p* = 0.001) interactions were observed for %HR_MAX_. Comparisons between sexes and competitive levels for changes in heart rate are illustrated in Figure 1A and Figure 1B, respectively. Moderate to large elevations in heart rate, compared to baseline, were occurred for men and Rx competitors prior to exercise on Workout 1 (*dz* = 2.46–2.98, *p* ≤ 0.005) and Workout 2 (*dz* = 1.40–1.75, *p* ≤ 0.027), but not for women or Scaled competitors. Among all competitors, similarly large elevations in heart rate from PRE were seen during each workout (*dz* = 8.78–11.33, *p* < 0.001) and they remained elevated (*dz* = 1.79–5.06, *p* ≤ 0.025) for 30-min post-exercise. Compared to Workout 1, Workout 2 elevations were greater throughout the entire recovery period for women and Rx competitors (*dz* = 0.84–2.22, *p* ≤ 0.012) but only between 5–15 min post-exercise for men (*dz* = 1.11–2.64, *p* ≤ 0.018) and between 10–25 min post-exercise for Scaled (*dz* = 0.72–1.84, *p* ≤ 0.021).

### 3.1. Anxiety

A significant main effect for time was observed for anxiety (F = 4.66, *p* ≤ 0.001). Compared to baseline (1.59 ± 0.25 au) and prior to Workout 1 (1.89 ± 0.33 au), moderately greater (*dz* = 0.80–1.04, *p* < 0.05) anxiety was seen prior to Workout 2 (2.85 ± 0.38 au). Compared to PRE, anxiety was decreased on both weeks at IP (Workout 1: 1.29 ± 0.12 au, *dz* = 0.71, *p* = 0.038; Workout 2: 1.93 ± 0.30 au, *dz* = 1.90, *p* = 0.002), 30P (Workout 1: 1.18 ± 0.11 au, *dz* = 0.85, *p* = 0.028; Workout 2: 1.41 ± 0.25 au, *dz* = 1.93, *p* < 0.001), and 60P (Workout 1: 1.18 ± 0.12 au, *dz* = 0.85, *p* = 0.035; Workout 2: 1.33 ± 0.27 au, *dz* = 1.90, *p* < 0.001). Anxiety in Workout 2 was higher at IP than Workout 1 though the effect was small (*dz* = 0.51, p = 0.038). No interactions or other specific differences were observed.

### 3.2. Catecholamine Response

No significant interactions were observed. Significant main effects for time were noted for epinephrine (F = 5.94, *p* < 0.001) and norepinephrine (F = 2.66, *p* = 0.040). Compared to PRE-concentrations, small to moderate elevations in epinephrine were seen at IP on both workouts (*dz* = 0.58–1.32, *p* ≤ 0.014) but not at other time points. However, epinephrine concentrations were moderately greater (*dz* = 1.06–1.33, *p* ≤ 0.014) in Workout 2 at each time point. Changes in norepinephrine concentrations only occurred in Workout 2, where compared to PRE, norepinephrine was moderately elevated at IP (*dz* = 0.92, *p* = 0.002) followed by a small depression at 60P (*dz* = 0.41, *p* = 0.014). Additionally, norepinephrine concentrations were moderately greater on Workout 2 at PRE (*dz* = 0.99, *p* = 0.002), IP (*dz* = 1.06, *p* = 0.020), and 30P (*dz* = 0.77, *p* = 0.037). No other differences were observed. Changes in salivary concentrations of epinephrine and norepinephrine are illustrated in Figure 2.

### 3.3. Heart Rate Variability

Although no significant interactions were observed, a significant main effect for time was noted for lnRMSSD (F = 18.84, *p* < 0.001). No differences were seen between baseline and PRE-recordings on either workout. Compared to their respective PRE-recordings, both workouts elicited large decreases in lnRMSSD at 5–10 min post-exercise (*dz* = 1.87–1.91, *p* < 0.001) and these remained depressed throughout the entire 30-min post-exercise recovery period (*dz* = 1.29–3.80, *p* < 0.001). Greater decreases were observed on Workout 2 at 20–30 min post-exercise (*dz* = 0.58–0.60, *p* ≤ 0.028). Between-workout comparisons for lnRMSSD are illustrated in Figure 3. 

Significant sex x time interactions were observed for lnHF (F = 2.66, *p* = 0.016), lnLF (F = 5.39, *p* < 0.001), and the LF:HF ratio (F = 2.35, *p* = 0.049). No differences were seen between baseline values for each of these variables and each workout’s respective PRE-values in men and women, except for a greater LF:HF ratio in men on Workout 2 (*dz* = 1.45, *p* = 0.028). Compared to each workout’s respective PRE-recordings, lnHF and lnLF were significantly (*dz* = 1.29–5.96, *p* < 0.05) depressed throughout the entire 30-min post-exercise recovery period for both sexes. However, lnHF was moderately lower in Workout 2 at 5–10 min post-exercise for all competitors (*dz* = 0.97, *p* ≤ 0.028), while lnLF was moderately lower in Workout 2 at PRE for men (*dz* = 1.22, *p* = 0.044) and at 5–10 min post-exercise for women (*dz* = 1.80, *p* = 0.022). The LF:HF ratio was only moderately elevated (*dz* = 1.02–1.46, *p* < 0.05) from its PRE-ratio in men on Workout 2 at 10–25 min post-exercise, with the ratio exceeding Workout 1 values at 20–25 min post-exercise (*dz* = 0.78, *p* = 0.032). Between-sex HRV comparisons are illustrated in Figure 4.

Significant competitive level x time interactions were observed for lnHF (F = 6.38, *p* < 0.001), lnLF (F = 6.17, *p* < 0.001), and the LF:HF ratio (F = 2.42, *p* = 0.045). In Rx competitors, baseline recordings were moderately lower than PRE-recordings for lnLF (Workout 1 only, *dz* = 8.82, *p* = 0.025) and for the LF:HF ratio (both workouts, *dz* = 1.21–1.45, *p* ≤ 0.017). Compared to PRE, lnHF and lnLF were depressed (*dz* = 1.14–4.29, *p* < 0.05) throughout the entire 30-min post-exercise period on both workouts, whereas the LF:HF ratio was only elevated (*dz* = 0.71–1.32, *p* < 0.05) in Workout 2 at 10–30 min post-exercise. Compared to Workout 1, lnHF was moderately lower at 5–10 min post-exercise (*dz* = 1.36, *p* = 0.002) and lnLF was lower at 10–15 min post-exercise (*dz* = 0.51, *p* = 0.016) in Workout 2. In scaled competitors, PRE-lnLF was lower than baseline recordings in Workout 1 (*dz* = 0.57, *p* = 0.029). In both workouts, lnHF and lnLF were significantly depressed (*dz* = 1.41–8.15, *p* < 0.05) from PRE-recordings throughout the entire 30-min post-exercise period except for lnLF at 25–30 min post-exercise on Workout 2 (*p* = 0.076). Compared to Workout 1, lnLF (at 25–30 min post-exercise) and the LF:HF ratio (at PRE and 15–20 min post-exercise) were greater in Workout 2 (dz = 1.10–7.26, *p* ≤ 0.039). Between-competitive level HRV comparisons are illustrated in Figure 5.

## 4. Discussion

The present study set out to examine anxiety and autonomic activity surrounding two consecutive HIFT competition workouts in recreational competitors. The specific workouts (i.e., the third and fourth workouts of the 2016 Open) varied in composition and duration, but their relative intensity (i.e., %HR_MAX_) was the same regardless of sex and competitive level. Nevertheless, differences were noted between workouts in anxiety, catecholamine concentrations, and HRV-derived indices surrounding exercise. All competitors reported greater anxiety prior to exercise in Workout 2 but similarly reduced anxiety post-exercise following each workout. Higher catecholamine concentrations were also observed in all competitors surrounding exercise in Workout 2, but these also returned to pre-exercise concentrations within 30 min of both workouts. In contrast, differing responses in HRV-derived indices surrounding exercise were noted between sexes and competitive levels for each workout. Previously, only one other study has ventured an investigation into aspects related HIFT competition recovery [26]. Using an indirect marker (i.e., the testosterone-to-cortisol ratio), the authors concluded that the Open might be considered an overreaching period within the context of the normal training for recreational competitors. The present study expands on those findings by demonstrating the influence of two consecutive HIFT competition workouts on anxiety and autonomic nervous system activity, and how sex and competitive level might affect their recovery.

Research on autonomic nervous system activity has predominantly occurred within the confines of the laboratory (i.e., a controlled, artificial environment) [6,15,27]. Previously, our lab compared autonomic nervous system activity surrounding various HIFT workouts and found no differences either at baseline or post-exercise [6,15]. Although collectively, these studies have provided the foundations of our current understanding of the autonomic nervous system response to stress and exercise, the training atmosphere within the laboratory does not emulate the competitive environment. The novelty of the current study was the onsite collection of data during HIFT competition. Here, we observed differences in pre-exercise anxiety and autonomic nervous system activity between workouts and compared to baseline, as well as between workouts following exercise. Because the competitive environment alters the psychological and physiological context of a workout, downstream mechanisms related to stress accommodation may also be altered. That is, participants individually complete required tasks within the laboratory [6,15] and rely on intrinsic motivational factors to perform well. In contrast, extrinsic factors (e.g., current and potential competitive ranking, the existence and performance of other competitors, limited options in modifying workouts or delaying their completion) may have influenced our participants’ perception of each workouts’ difficulty and importance, as well as their effort ability to recover. The added stress of competitive sports and its nature (i.e., setting, opponent quality, provocation), compared to normal training, is known to impact pre-competition anxiety and impact the autonomic nervous system response to exercise [14,16,17]. Our findings support this hypothesis and suggest that autonomic nervous system function surrounding HIFT competition will respond differently than what occurs within the controlled laboratory environment.

During competitive workouts, time-dependent changes in anxiety, catecholamine concentrations, and HRV-derived indices were noted and expected. Previously, increases in circulating catecholamines and decreases in HRV-derived indices had been documented following the benchmark workouts “Cindy” [6], “Grace” [15], and a 15-min AMRAP [15] before returning to pre-exercise levels within 1–2 h [6,15], with no differences being seen between HIFT protocols [15]. Elevated catecholamines and depressed HRV, along with reduced anxiety, were also seen following both workouts of the present study but their magnitude was greater on Workout 2. While the return of anxiety, epinephrine, and norepinephrine to pre-exercise values within 30 min of completing both workouts was consistent with previous reports [6,15], the lack of HRV recovery was not and appeared to be modulated by the specific workout and the athletes’ competitive level (i.e., there was evidence of recovery in scaled competitors but not Rx). Despite relative intensity (i.e., %HR_MAX_) being comparable between workouts, duration (7 min vs. 13 min), expected volume (195–225 repetitions vs. 660 repetitions), and exercise complexity were different across competitive levels (see Table 1). Exercise complexity is particularly of interest here because competitors were given the option of selecting the workout option that would enable them to earn their highest score. Unfortunately, their choice was limited by their ability to perform specific tasks or lift the required load. For example, a competitor would have had to have completed the scaled option if they could not perform the bar muscle-up requirement of Workout 1 or lift the Rx load prescribed for the deadlift (255 lbs. for men; 155 lbs. for women) in Workout 2. Unlike previous reports [6,15], workout design and possibly competitive level appear to impact autonomic nervous system activity surrounding HIFT exercise. Moreover, it is worth noting that the specific observed responses may also have been modulated by our participants’ self-selected warm-up. The sophistication of a warm-up is known to affect an athlete’s readiness (physiological and mental) to compete [28]. Since our foreknowledge of each workout’s programming was limited and because programming was slightly different for each sex and competitive level (i.e., Rx and Scaled), we did not use a standard warm-up prior to each workout. Although we did not observe any extraordinary differences between each participants’ self-selected warm-up protocol, slight differences may have influenced our findings.

The lack of uniformity between subjective anxiety, catecholamine concentrations, and HRV responses is also interesting. Anxiety and catecholamine concentrations were all greater prior to exercise in Workout 2, significantly responded to both workouts, and were recovered within 30 min. In contrast, HRV-derived indices were generally similar at PRE and immediately following exercise, and they did not recover within 30 min. This poses a compelling hypothesis that anxiety may provide a stronger stimulus to the sympathetic branch than the parasympathetic. This was an unexpected observation because autonomic nervous system balance occurs between the parasympathetic and sympathetic nervous systems with close to a “zero-sum-gain”. Nevertheless, the non-uniform responses from parasympathetic markers (lnRMSSD and lnHF) compared to the other measures suggest that resting vagal inhibition likely overshadows the effects of circulating epinephrine and norepinephrine. Heightened pre-exercise sympathetic activity may inhibit vagal rebound and possibly interfere with the relationship between these measures during recovery. It is possible that several of the previously mentioned extrinsic motivational factors related to the competition (e.g., the importance of the upcoming workout on competitive rank), and possibly each participant’s strategy for warming up, may have affected pre-exercise sympathetic activity. To better determine the influence of a heightened pre-exercise sympathetic response, future studies might attempt to control for potential extrinsic motivational factors (e.g., competitive rank) and implement a standardized warm-up protocol.

## 5. Conclusions

The primary purpose of this study was to evaluate the feasibility of measuring autonomic nervous system recovery following two different Open workouts within a competitive environment. Within this context, however, it is difficult to extrapolate the full magnitude of the delayed autonomic nervous system rebound observed following both exercise bouts. Unlike previous reports on HIFT performed in a laboratory setting [6,15], differences were noted between measures collected during the baseline visit and pre-exercise, as well between workouts at PRE. The competitors reported being more anxious and were found to have higher catecholamine concentrations prior to exercise in Workout 2, whereas the differences noted in some HRV-derived indices appeared to be dependent on the workout and their sex and competitive level. Differences between workouts were also noted post-exercise. Although anxiety and catecholamine concentrations were recovered within 30 min of exercise, values were higher throughout the entire 60-min recovery period in Workout 2. In contrast, HRV measures did not recover within 30 min of exercise, but, with certain exceptions, were generally consistent with baseline values prior to exercise each week. Collectively, we might conclude that our participants were sufficiently recovered prior to exercise on both weeks. However, the lack of uniformity between subjective anxiety, catecholamine concentrations, and HRV-derived indices surrounding exercise demonstrate the need for more onsite data collection. Our data implies that the relationships between these measures and their response to HIFT, particularly HIFT competition, may not be fully captured in the laboratory. Future investigations may expand on our findings by investigating autonomic nervous system activity, as well as other markers indicative of recovery, in relation to HIFT competition in both Rx and scaled competitors. Additional factors such as training volume throughout this competitive period, utilizing multiple attempts on weekly challenges, and examining autonomic nervous system activity over a longer post-exercise recovery period may also help to provide insight. In the meantime, researchers and practitioners might view our findings as preliminary evidence of recovery between two consecutive, weekly competitive bouts during the Open in recreational competitors.

## Figures and Tables

**Figure 1 sports-07-00199-f001:**
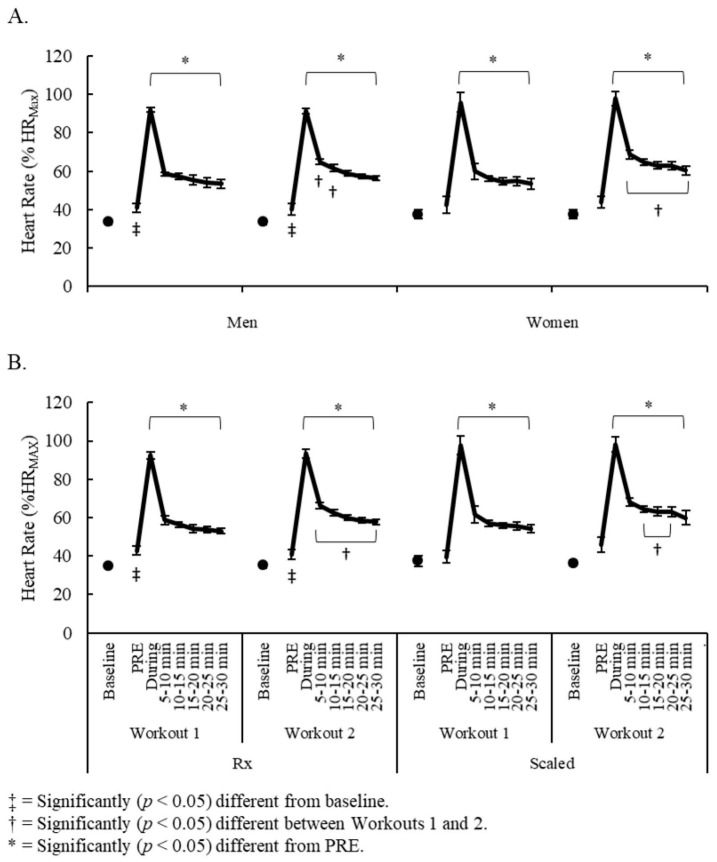
Changes in heart rate comparisons between (**A**) men and women; and (**B**) Rx and Scaled competitors.

**Figure 2 sports-07-00199-f002:**
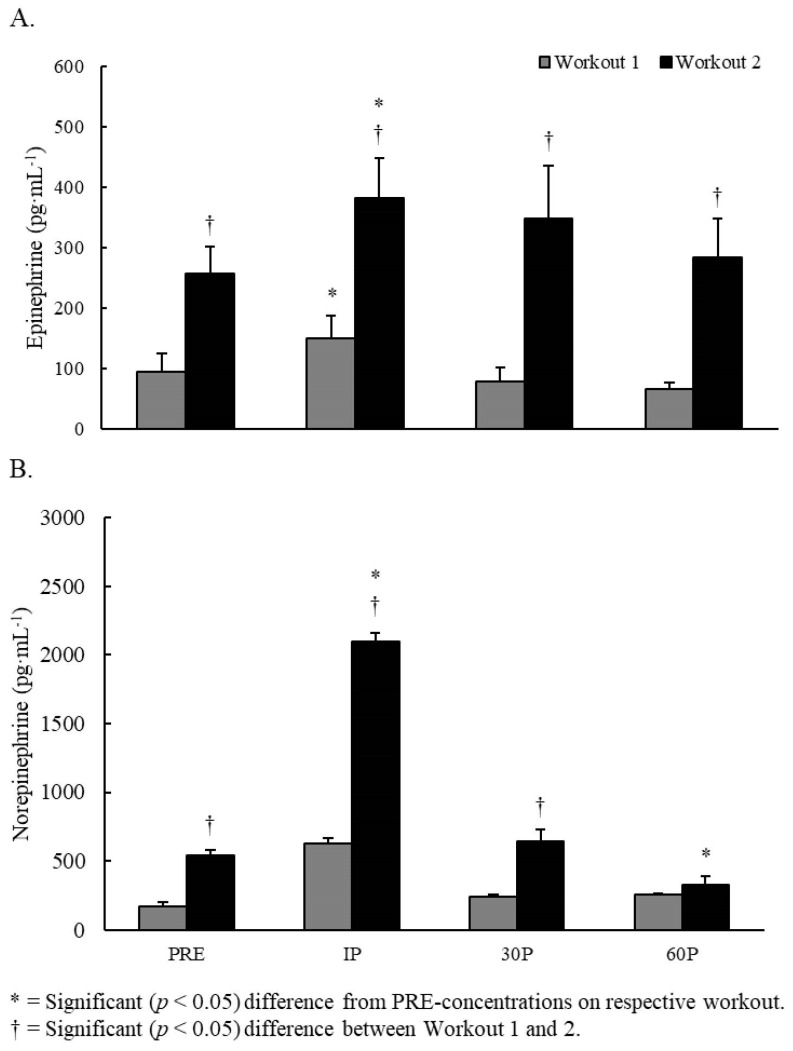
Changes in saliva concentrations of (**A**) Epinephrine and (**B**) Norepinephrine.

**Figure 3 sports-07-00199-f003:**
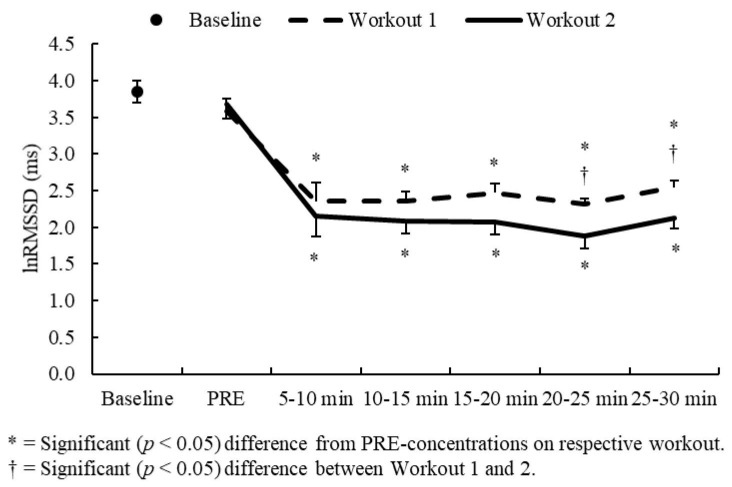
lnRMSS comparisons between Workouts 1 and Workout 2.

**Figure 4 sports-07-00199-f004:**
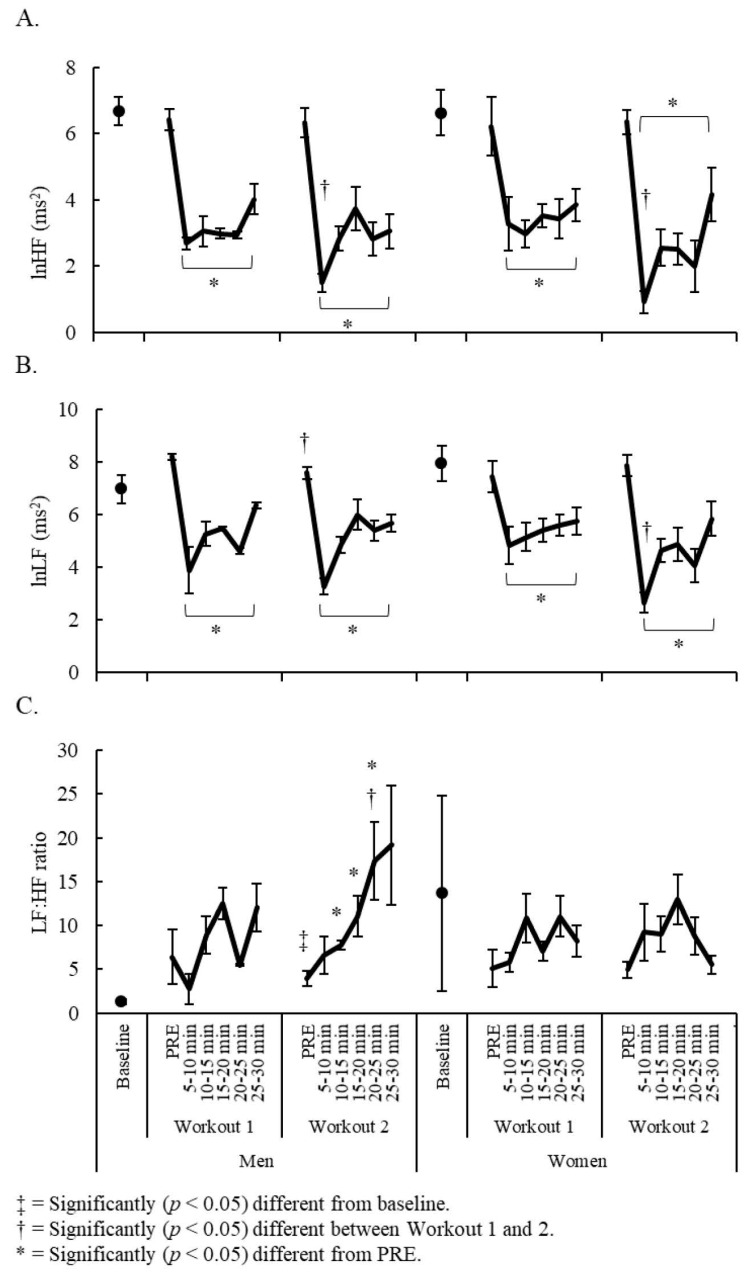
Sex and week comparisons for (**A**) lnHF; (**B**) lnLF; and (**C**) LF:HF ratio.

**Figure 5 sports-07-00199-f005:**
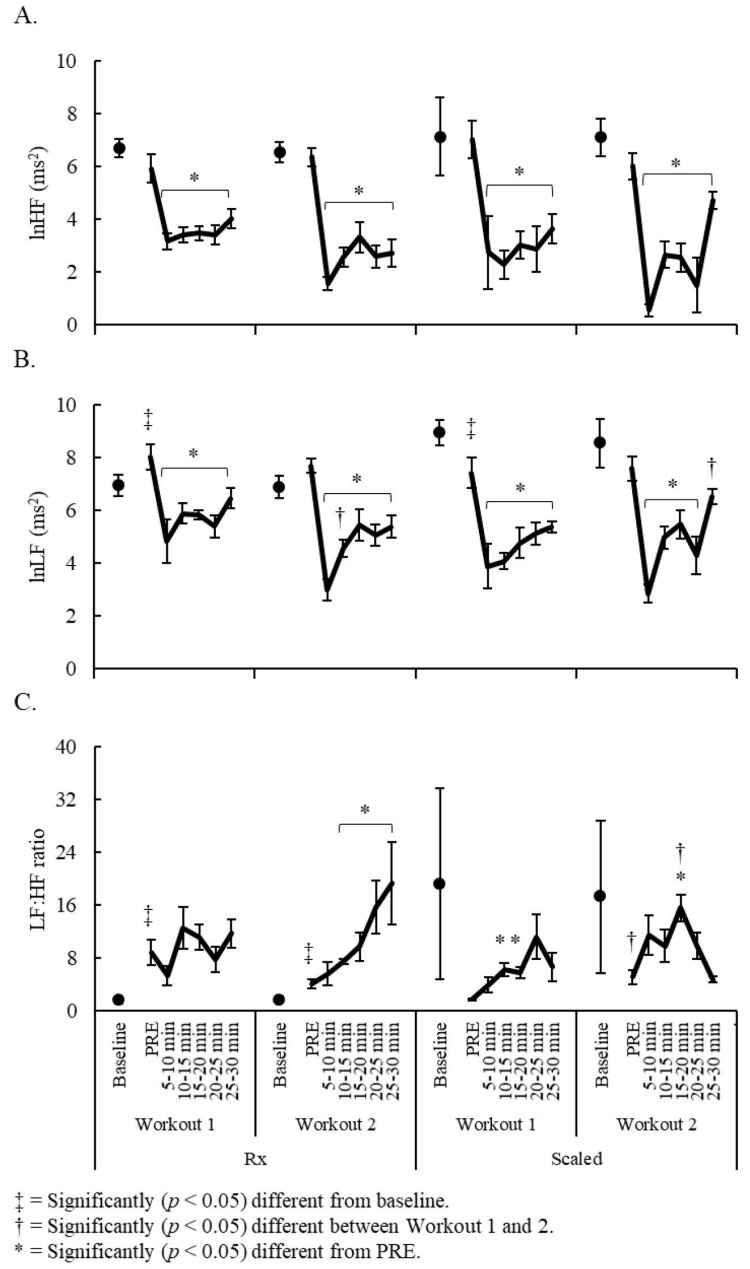
Competitive level and week comparisons for (**A**) lnHF; (**B**) lnLF; and (**C**) LF:HF ratio.

**Table 1 sports-07-00199-t001:** Workout descriptions and performance scores for (male / female) competitors of the 2016 Open (mean ± standard deviation).

	Rx	Scaled
Workout 1	7-min AMRAP	7-min AMRAP
Type		
Exercises	10 power snatches (75 / 55 lbs.) 3 bar muscle-ups	10 power snatches (45 / 35 lbs.) 5 jumping chest-to-bar pull-ups
Max Score *	420 repetitions (195 repetitions)	225 repetitions
Actual Score		
Men	275 ± 9 repetitions (80 ± 9 repetitions)	-
Women	229 ± 34 repetitions (34 ± 34 repetitions)	100 ± 48 repetitions
Total	262 ± 27 repetitions (67 ± 27 repetitions)	100 ± 48 repetitions
Workout 2	13-min AMRAP	13-min AMRAP
Type		
Exercises	55 deadlifts (225 / 155 lbs.) 55 wall-ball shots (20 / 14 lbs. to 10 / 9 ft. target) 55 calories on rowing ergometer 55 handstand push-ups	55 deadlifts (135 / 95 lbs.) 55 wall-ball shots (20 /10 lbs. to 9 / 9 ft. target) 55 calories on rowing ergometer 55 hand-release push-ups
Max Score *	1320 repetitions (660 repetitions)	660 repetitions
Actual Score		
Men	847 ± 9 repetitions (187 ± 9 repetitions)	-
Women	825 repetitions (165 repetitions)	195 ± 68 repetitions
Total	844 ± 12 repetitions (184 ± 12 repetitions)	195 ± 68 repetitions

*Note*: AMRAP = As many repetitions as possible; Rx = As prescribed. * Rx competitors are always ranked above scaled competitors, regardless of score. Thus, max score for Rx competitors is indicated by the number of repetitions completed in addition to the maximum possible scaled score. The actual number of repetitions completed is indicated within parentheses.

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
