# Peer review of "Pre-Anticipatory Anxiety and Autonomic Nervous System Response to Two Unique Fitness Competition Workouts"

_sports, 2019, doi:10.3390/sports7090199_

Round 1
Reviewer 1 Report
Pre-anticipatory anxiety and autonomic nervous system response to two unique fitness competition workouts
This is an interesting manuscript from Mangine and colleagues looking at various autonomic responses to CrossFit workouts. I think the manuscript is interesting and the work is timely. There are aspects that could have been better controlled such as having a standardised warm up before the competition. However, these are the inherent limitations of working with athletes. The manuscript needs significant revisions to improve clarity, mainly fully-spelling most abbreviated terms and re-labelling of figures. Once this is complete, I suggest the manuscript will be suitable for acceptance as it would be of value to the strength and conditioning field, and those working with athletes in this domain.
Major comments
· Reduce the amount of abbreviations. It is unnecessary and the sheer volume of abbreviations makes the manuscript difficult to read e.g. E, NE, ANX, IP, P, OFC, PNS, ANS, HF, LF. Where abbreviations are standard nomenclature e.g. HRV then this is fair enough, and perhaps for the odd term like HIFT where spelt out would make the text clunky. Other than this, I would spell everything out; it will improve the flow of the manuscript greatly and make it far easier for readers that want specific info and don’t want to read the entire manuscript.
· Line 134-140 – I realise this is a limitation of working with athletes, but a standardised warm-up would increase the confidence in the results greatly. This should be touched upon in the discussion
Minor comments
· Line 34, worth defining that you are talking about CrossFit
· Line 43-53. I don’t disagree with most of these data around oxidative stress and inflammation, but I think this needs to be more related to function. You also don't measure inflammation or oxidative stress. Athletes mainly care about fatigue, loss of muscle power, endurance etc. Having elevated levels of ROS or TNF will mean little or nothing to most athletes and as researchers, we still don’t fully understand the time-course of these processes in how they relate to training, overtraining, recovery, and athletic performance. Could you condense this and add a focus on more functional outcomes?
· Line 67-68 please could you give a brief description of what these workouts entail e.g. are they running-based, Olympic lifting based? Just to give an indication to someone not familiar with these workouts
· Line 81 please give more details on the University and which board approved the study i.e. was it the ethics board of a specific faculty?
· The females in the study seem to have a high body mass for CrossFit athletes; they have an average BMI of 30.5. Is this correct – were any overweight? Were they particularly from a powerlifting background? Do you have any data on body fat % from the Tanita? It may be worth adding further clarity to your participant group to address this point.
· Heart rate monitor details line 95?
· Line 163 – reference for age predicted max? You could use ACSM or similar perhaps?
· Line 188-197 the way stats are reported is a little confusing. We use standard P values such as <0.05 (5%); <0.01 (1%); <0.001 (0.1%); or <0.0001 (0.01%) to report on the chance we are incorrectly rejecting a null hypothesis. If you would like to report exact P values, this is fine, but I would report as P = 0.027, for example.
· Line 199-205 the stats are reported in this manner – is there a reason for this that I’ve missed?
· Line 211 – 218 – is it necessary to use W3/W4 or could you just say ‘week 3, week 4’? Sentences like 215-216: ‘Additionally, NE concentrations were greater on W4 at PRE (p = 0.002), IP (p = 0.020), and 30P (p = 0.037)’ is not very clear because of the volume of abbreviations.
· Line 218 you use epinephrine and norepinephrine, despite abbreviating elsewhere. Spell in full throughout like this, please.
· On figures, the abbreviations used (as above) are confusing. PRE seems self-explanatory but IP, 30P and 60P I had to double check back to when they were first referenced, could these be spelt in full to aid clarity, please?
· Line 230-53 what are LF and HF?
· With the heart rate graphs, what is the rationale for using %HR max? Your groups are well matched for age so I would be inclined to used absolute heart rate beats per min.
· Line 219 – looking at the figures calling the workouts ‘week 3’ and ‘week 4’ is confusing as I immediately wonder where the data are for weeks 1 and 2. Could these not just be named as the workouts? You can clarify the above in the methods section, so the details are available if it is needed but week ¾ means nothing to a reader whereas if it said X workout vs Y workout it’s immediately clear. Same goes for all figures – scrap the week 3 or 4 and just name the workouts
· Conclusions can be refined to focus on the results of this study. Line 336 suggests that data have been split to produce two manuscripts but it would have been far better to include all in the one script so that they can be considered alongside each other. The reference is incorrect on this statement too – presumably you mean reference 20?
I would suggest going through all references to ensure they are accurately referencing what you say that they are.
Author Response
This is an interesting manuscript from Mangine and colleagues looking at various autonomic responses to CrossFit workouts. I think the manuscript is interesting and the work is timely. There are aspects that could have been better controlled such as having a standardised warm up before the competition. However, these are the inherent limitations of working with athletes. The manuscript needs significant revisions to improve clarity, mainly fully-spelling most abbreviated terms and re-labelling of figures. Once this is complete, I suggest the manuscript will be suitable for acceptance as it would be of value to the strength and conditioning field, and those working with athletes in this domain.
Thank you for your excellent comments and suggestions, and for your thorough review. We have addressed each item here and throughout the manuscript. We believe that you will find that the changes made to this manuscript will satisfy your previous reservations and warrant its publication in Sports.
Major Comments:
Reduce the amount of abbreviations. It is unnecessary and the sheer volume of abbreviations makes the manuscript difficult to read e.g. E, NE, ANX, IP, P, OFC, PNS, ANS, HF, LF. Where abbreviations are standard nomenclature e.g. HRV then this is fair enough, and perhaps for the odd term like HIFT where spelt out would make the text clunky. Other than this, I would spell everything out; it will improve the flow of the manuscript greatly and make it far easier for readers that want specific info and don’t want to read the entire manuscript.
We have reduced the number of terms that were abbreviated, and now spell out the following terms: Epinephrine, Norepinephrine, Anxiety, Online Fitness Competition, Parasympathetic Nervous System, Sympathetic Nervous System, and Autonomic Nervous System. However, for brevity in text and figure presentation, we would like to keep HF, LF, RMSSD, PRE, IP, 30P, and 60P, as well as HIFT.
Line 134-140 – I realise this is a limitation of working with athletes, but a standardised warm-up would increase the confidence in the results greatly. This should be touched upon in the discussion
We acknowledge that the warm-up strategy can influence performance. Although we did not document each participant’s specific warm-up, we did not observe any practices that seemed unusual. Typically, each participant’s warm-up included light aerobic exercise, stretching, and exercise (i.e., those to be performed in each workout) practice. Moreover, we did not observe participants using loads of greater intensity (i.e., attempting to elicit a potentiation effect) than what would be required in the workout, nor did we observe any participant performing an alarming number of repetitions during the warm-up that would elicit enough fatigue to compromise performance. While we agree that a standardized warm-up would have been ideal, there are reasons for why this would not have been practical. First, the timeline was extremely limited. Like the participants, we were not made aware of each workout’s details until the night before many of our participants made their first attempt. It would have been very difficult to devise an appropriate, standardized warm-up that was specific to each workout while also coordinating any necessary changes to data collection procedures. For instance, upon each workout’s release, we immediately began to determine an appropriate number of available research team members for each participant to ensure repetition count and quality (coordinating with judges) and identifying potential workout strategies that would require monitoring. A second reason is that a standardized warm-up that was unusual for a participant could have also impacted performance. It is not uncommon for researchers to allow participants to perform additional preparatory exercises following a standard general warm-up before maximal testing. Since we were monitoring autonomic function, we felt that allowing participants to complete a warm-up of their choice would avoid creating an additional stress and be more consistent with our primary aims (i.e., to observe the autonomic response in a competitive environment).
We have acknowledged the potential influence of our decision on our results within our discussion.
Minor Comments:
Line 34, worth defining that you are talking about CrossFit
Thank you for this comment. We had avoided using this specific term in our original submission (other than in our Key Words) in attempt to limit any pre-conceived notions our readers might have on CrossFit; notions that are predominantly mixed between in-favor or heavily against. Nevertheless, we agree that it is more suitable to identify the online fitness competition by its true name.
The following was added: The most popular competition, featuring high-intensity functional training (HIFT), is an annual, international, multi-stage event used to identify ‘the fittest on earth’ held by CrossFit®. We have also amended terminology throughout the entire manuscript.
Line 43-53. I don’t disagree with most of these data around oxidative stress and inflammation, but I think this needs to be more related to function. You also don't measure inflammation or oxidative stress. Athletes mainly care about fatigue, loss of muscle power, endurance etc. Having elevated levels of ROS or TNF will mean little or nothing to most athletes and as researchers, we still don’t fully understand the time-course of these processes in how they relate to training, overtraining, recovery, and athletic performance. Could you condense this and add a focus on more functional outcomes?
Yes, we can. Prior to our original submission, we had some discussion on whether markers of inflammation and oxidative stress were needed in this paragraph. Since research on HIFT is so limited and these markers are often included in the discussion on recovery, regardless of whether they are important, we felt compelled to include them. That said, Tibana et al. (2016) supplemented their biochemical markers with an assessment of back squat power following two consecutive days of HIFT. Although this specific exercise may have lacked some relevance to the actual daily workouts, their findings are intriguing. Therefore, we have amended this section to limit discussion on biomarkers of inflammation and oxidative stress and added a brief discussion on power.
Line 67-68 please could you give a brief description of what these workouts entail e.g. are they running-based, Olympic lifting based? Just to give an indication to someone not familiar with these workouts
We have added brief descriptions of the exercises included in these workouts.
Line 81 please give more details on the University and which board approved the study i.e. was it the ethics board of a specific faculty?
At Kennesaw State University, our Institutional Review Board (IRB) is the only committee that oversees the safety and ethics of the project. We have revised this statement to identify the specific IRB and in accordance with the author guidelines set forth by Sports.
The females in the study seem to have a high body mass for CrossFit athletes; they have an average BMI of 30.5. Is this correct – were any overweight? Were they particularly from a powerlifting background? Do you have any data on body fat % from the Tanita? It may be worth adding further clarity to your participant group to address this point.
We did assess body fat percentage via dual X-ray absorptiometry (DXA) at the beginning of the participants’ baseline visit but were hesitant to report this information in our original submission. The relevance of body composition to the present investigation is limited to being only a descriptor and DXA alone does not adequately describe composition. Since our study included recreational athletes, who represent a very large and diverse portion of the Open competitor population, it is reasonable to expect that their body composition would not resemble that of competitors who advance to later rounds of the competition. Nevertheless, we acknowledge the benefit of providing this information to readers and researchers for comparative purposes in future investigations and have amended our manuscript to detail its measurement and results. However, we are not comfortable with using these data (or BMI) to classify any of the participants as being overweight or obese. BMI is typically higher in athletic populations and does not consider the contribution of lean tissue (muscular or skeletal) on mass. While DXA does provide these distinctions, it is primarily designed to measure skeletal mass and then estimates muscle and fat tissue based on absorbance. A more accurate method would be to combine the results of several assessment techniques (i.e., 4 – 6 compartment models) (Wang et al. 1998) but this was not a primary aim of the investigation and appropriate steps towards accurate measurement (i.e., fasted state, multiple measurements, etc.) were not taken. Therefore, we have added body composition information, obtained via DXA, to this paper with extreme caution.
Wang Z, Deurenberg P, Guo S, Pietrobelli A, Wang J, Pierson Jr RN, and Heymsfield S. Six-compartment body composition model: Inter-method comparisons of total body fat measurement. International Journal of Obesity, 22(4), 329-337, 1998
Heart rate monitor details line 95?
The following was added: Upon arrival, participants were fitted with a polar Team2 heart rate (HR) monitor (Lake Success, NY)
Line 163 – reference for age predicted max? You could use ACSM or similar perhaps?
We have added the following reference:
Gibson, A.L.; Wagner, D.R.; Heyward, V.H. Designing cardiorespiratory exercise programs. In Advanced fitness assessment and exercise prescription, 8th ed.; Human Kinetics: Champaign, IL, 2019; pp 125 - 157
Line 188-197 the way stats are reported is a little confusing. We use standard P values such as <0.05 (5%); <0.01 (1%); <0.001 (0.1%); or <0.0001 (0.01%) to report on the chance we are incorrectly rejecting a null hypothesis. If you would like to report exact P values, this is fine, but I would report as P = 0.027, for example.
Line 199-205 the stats are reported in this manner – is there a reason for this that I’ve missed?
We apologize for the confusion regarding our manner of reporting some p-values, but we felt that doing so would be more informative to the reader. In your example,
“Heart rate was elevated from baseline for men and Rx competitors at W3-PRE (p ≤ 0.005) and at W4-PRE (p ≤ 0.027), but not for women or Scaled competitors.”
We are referencing multiple p-values. That is, compared to baseline, the p-value for men and the p-value for Rx competitors were both less than or equal to 0.005 on W3-PRE and 0.027 on W4-PRE. It would be less informative to simply write p < 0.05 and writing out the p-value for each comparison would be redundant (i.e., several consecutive sentences describing similar comparisons) and possibly more confusing. Therefore, we would prefer to continue to report these values in this manner, but we are willing to accommodate your request if you feel that it is necessary.
Line 211 – 218 – is it necessary to use W3/W4 or could you just say ‘week 3, week 4’? Sentences like 215-216: ‘Additionally, NE concentrations were greater on W4 at PRE (p = 0.002), IP (p = 0.020), and 30P (p = 0.037)’ is not very clear because of the volume of abbreviations.
Many of the abbreviations have been removed and we now refer to week 3 and week4 as Workouts 1 and 2, according to your later comment.
Line 218 you use epinephrine and norepinephrine, despite abbreviating elsewhere. Spell in full throughout like this, please.
We have revised the manuscript throughout to spell out epinephrine and norepinephrine.
On figures, the abbreviations used (as above) are confusing. PRE seems self-explanatory but IP, 30P and 60P I had to double check back to when they were first referenced, could these be spelt in full to aid clarity, please?
It is common to use this abbreviation strategy for post-exercise time points (i.e., IP = immediately post, 30P = 30-minutes post, and 60P = 60-minutes post) for brevity (see referenced articles). It also helps to avoid text overlap in figures and tables. However, we have limited our usage of these abbreviations to aid in clarity.
Hoffman et al. "Effect of a pre-exercise energy supplement on the acute hormonal response to resistance exercise." The Journal of Strength & Conditioning Research 22.3 (2008): 874-882. Mangine et al. "The effect of training volume and intensity on improvements in muscular strength and size in resistance‐trained men." Physiological reports 3.8 (2015): e12472. Townsend et al. "Effects of β-hydroxy-β-methylbutyrate free acid ingestion and resistance exercise on the acute endocrine response." International journal of endocrinology 2015 (2015).
Line 230-53 what are LF and HF?
These are common nomenclature for Low Frequency Domain (LF) and High Frequency Domain (HF) in HRV. These are defined in the methods section for reference.
With the heart rate graphs, what is the rationale for using %HR max? Your groups are well matched for age so I would be inclined to used absolute heart rate beats per min.
This is a great comment. We chose to use %HR max to “normalize” the data amongst the participants and to provide a rough estimation of exercise intensity. While we would usually agree that using absolute or raw values is a better practice, it is not common for heart rate in HIFT literature (see referenced articles). Therefore, for uniformity and to enable easer comparisons when reading through the HIFT literature, we chose to use percentage of maximal heart rate. It is worth noting that similar statistical differences were noted when we had examined absolute values.
Babiash PE. Determining the energy expenditure and relative intensity of two crossfit workouts. University of Wisconsin-La Crosse, 2013. Feito Y, Moriarty TA, Mangine G, and Monahan J. The use of a smart-textile garment during high-intensity functional training. A Pilot Study. The Journal of sports medicine and physical fitness. 2018. Fernández JF, Solana RS, Moya D, Marin JMS, and Ramón MM. Acute physiological responses during crossfit® workouts. European Journal of Human Movement. 35: 114-24, 2015 Kliszczewicz B, Snarr R, and Esco M. Metabolic and Cardiovascular Response to the CrossFit Workout 'Cindy': A Pilot Study. J Sport Performance. 2 (2): 1-9, 2014 Kliszczewicz B, Buresh R, Bechke E, and Williamson C. Metabolic biomarkers following a short and long bout of high-intensity functional training in recreationally trained men. Journal of Human Sport and Exercise. 12 (3), 2017 Fernández JF, Solana RS, Moya D, Marin JMS, and Ramón MM. Acute physiological responses during crossfit® workouts. European Journal of Human Movement. 35: 114-24, 2015 Maté-Muñoz JL, Lougedo JH, Barba M, Cañuelo-Márquez AM, Guodemar-Pérez J, García-Fernández P, Lozano-Estevan MD, Alonso-Melero R, Sánchez-Calabuig MA, Ruíz-López M, and de Jesús F. Cardiometabolic and Muscular Fatigue Responses to Different CrossFit® Workouts. Journal of Sports Science and Medicine. 17 (4): 668-679, 2018 Tibana R, de Sousa N, Prestes J, and Voltarelli F. Lactate, Heart Rate and Rating of Perceived Exertion Responses to Shorter and Longer Duration CrossFit® Training Sessions. Journal of Functional Morphology and Kinesiology. 3 (4): 60, 2018.
Line 219 – looking at the figures calling the workouts ‘week 3’ and ‘week 4’ is confusing as I immediately wonder where the data are for weeks 1 and 2. Could these not just be named as the workouts? You can clarify the above in the methods section, so the details are available if it is needed but week ¾ means nothing to a reader whereas if it said X workout vs Y workout it’s immediately clear. Same goes for all figures – scrap the week 3 or 4 and just name the workouts
We agree and have revised the manuscript to refer to weeks 3 and 4 as workouts 1 and 2, respectively.
Conclusions can be refined to focus on the results of this study. Line 336 suggests that data have been split to produce two manuscripts but it would have been far better to include all in the one script so that they can be considered alongside each other. The reference is incorrect on this statement too – presumably you mean reference 20?
We have revised our discussion and conclusion to be more concise and focused on our main findings. We agree that it would have been more comprehensive to have included the present study’s findings with our previous publication. However, it was not possible to analyze our samples for catecholamine concentrations at the time the first manuscript was being prepared due to available funding. We also felt that the research questions were different enough to warrant two separate publications. In our first manuscript, we were primarily interested in describing the testosterone and cortisol responses to each workout. At the time, no other study had investigated their response to HIFT and so that study was very much exploratory in nature. Further, because no other study had examined these hormones in relation to HIFT, our comparisons were limited to what had been previously documented in relation to resistance training and other sports. In contrast, the present study set out to expand on previous reports on the catecholamine and HRV responses to HIFT. Although this study was also exploratory in nature, the ability to compare the data we obtained in the competitive setting with our baseline recordings (saliva samples excluded) and with what had been observed in a laboratory setting provided a unique opportunity. If we were to have included everything into one paper, it would likely have required a much longer discussion and possibly, confusion.
The reviewer was correct on the reference. However, due to our revision of the conclusion, it was no longer needed here.

Reviewer 2 Report
The authors have done a good job in collecting data in an applied area trying to conduct a complex analysis. The research question has value. However, four reasons effect the scientific contribution the paper makes.
a) the case for doing study and the central message from the study appear to change. The main finding related to gender was not made persuasively in the introduction.
b) a limitation with the introduction is that arguments are not persuasively made to establish hypotheses.
c) data analysis needed to include effect sizes and the percentage changes. The data analysis strategy needed to link to hypotheses. Therefore, the exploratory nature of the study needs recognising.
d) the acknowledged absence of control data means that it is possible to interpret data in many different ways.
I feel there is enough data for a worthwhile article. The authors need theory-led hypotheses and keep to those hypotheses. The discussion drifts into new areas, much of which is speculation.
Author Response
The authors have done a good job in collecting data in an applied area trying to conduct a complex analysis. The research question has value. However, four reasons effect the scientific contribution the paper makes.
Thank you for reviewing our manuscript. We have responded to each of your comments and amended the manuscript where applicable. We believe these clarifications provide satisfactory evidence of the scientific merit for this paper.
1. the case for doing study and the central message from the study appear to change. The main finding related to gender was not made persuasively in the introduction.
The rationale for adding gender, and competitive status for that matter, to the analyses was based on workout programming differences and explained in our statistical analysis section. Explanation in our Introduction was not warranted because we did not have a reason to suspect that autonomic function would be different between men and women due to physiological differences. Rather, programming (e.g., barbell and wall-ball loads, wall-ball target distance) is arbitrarily prescribed differently for men and women, as well as for Rx and Scaled. The exact prescription is not made relative to the athlete; it is simply the nature of the sport. Therefore, to account for the potential differences in the autonomic response brought on by difference in programming, we added sex and competitive level to the analysis. It would not have been possible to account for these differences in any other way.
2. a limitation with the introduction is that arguments are not persuasively made to establish hypotheses.
The limitations (or lack of persuasion) in our Introduction’s argument, to which you are referring, are not clear. Our Introduction explains that the competition occurs over five weeks and competitors will typically participate in their normal training habits during this period. Thus, recovery between competitive events is important. We go on to discuss the findings of literature that has investigated aspects of recovery in relation to HIFT. A major limitation with these few studies is that they have all been conducted in a controlled, laboratory setting. This setting does not emulate the typical competition environment, which is known to affect autonomic nervous system function. Therefore, our purpose was to examine the autonomic response and its recovery during consecutive weeks of the competition. No other study has attempted to examine autonomic function in this setting.
We believe that our argument’s limitations, to which you are referring, are related to our failure to include a statement in our Introduction regarding the effect of the competitive environment. We realized that we had pointed this out in our Discussion but not in our Introduction. We have amended this section accordingly.
3. data analysis needed to include effect sizes and the percentage changes. The data analysis strategy needed to link to hypotheses. Therefore, the exploratory nature of the study needs recognising.
We have indicated the exploratory nature of our study and added effect sizes to the analysis. However, we disagree that percent changes should have been used in our statistical analysis as it is more appropriate to analyze actual values whenever possible. By calculating the percent change between a pair of values, their context is lost. Changes that are small (in absolute terms) may appear inflated when the initial value is small, and thus, could increase the risk of committing Type 1 error.
4. the acknowledged absence of control data means that it is possible to interpret data in many different ways.
The within-design of this study enables participants to serve as their own controls. That is, our observations at baseline and then prior to exercise act as the controlled conditions to which comparisons are made post-exercise or between workouts.
Reviewer 3 Report
Modify abstract presenting a concrete objetive as well as conclusion
Do not use abbreviature for anxiety.
Explain why do not use validated test to analyse anxiety response Calculate %RHmax with more recent equation: http://www.onlinejacc.org/content/37/1/153.abstract Specific
literatura in hiit would improve the discussion of your data: https://link.springer.com/article/10.1007/s10916-017-0741-4
provide a conclusion that specifically responds to the objectives of the study.
Author Response
Thank you for your comments and suggestions. We have addressed each item and made appropriate revisions throughout the manuscript. We believe these changes have improved the manuscripts quality and warrant its publication in Sports.
Modify abstract presenting a concrete objective as well as conclusion
Within the journal’s word count limitations, we have revised our introductory and concluding statements to offer a more concrete objective and conclusion.
Do not use abbreviature for anxiety.
ANX has been changed to anxiety throughout the entire manuscript.
Explain why do not use validated test to analyse anxiety response
The validity and reliability of a single-item Likert scale to assess current anxiety has previously been shown to be valid and reliable (Davey et al. 2007). We have included a reference for this in our manuscript.
Davey et al. "A one-item question with a Likert or Visual Analog Scale adequately measured current anxiety." Journal of clinical epidemiology 60.4 (2007): 356-360.
Calculate %RHmax with more recent equation: http://www.onlinejacc.org/content/37/1/153.abstract Specific
We appreciate this suggestion but feel that the 220-age formula is sufficient for our study. While the Tanaka formula may be more recent and possibly more specific than Fox and Haskell’s equation, it is not necessarily more accurate or important to the present investigation. One point to consider is that Tanaka et al. advertises (in their Discussion) that their derived equation is more accurate. That 220-age over- and under-estimates maximal heart rate in younger and older populations, respectively. However, these claims are solely based on a comparison to their specific population and derived equation (i.e., two different samples were used to develop these regression equations). Moreover, Tanaka et al. produces their equation, reports variance explained by age, but never identifies the standard error of the estimate (SEE). It is odd that they do not report SEE because it would probably provide the strongest evidence for their equation’s accuracy. At least this information is available for the 220-age formula. Then when you consider that our purpose for calculating percentage of maximal heart rate was to provide a rough estimation of exercise intensity, and 220-age is commonly-used for this purpose, it seems more prudent and consistent with the literature to use that equation.
literatura in hiit would improve the discussion of your data: https://link.springer.com/article/10.1007/s10916-017-0741-4
We appreciate your suggesting the article by Clemente-Suárez and Arroyo-Toledo, but do not feel that it would be appropriate for making comparisons. Although the researchers also monitored performance and recovery with HRV, the context was quite different. For instance, the post-exercise recovery periods were either much different than ours. The authors used a 3-minute recovery period between tethered and HIIT session, and then used an active recovery session (i.e., 200-m recovery swim) following the HIIT session. In contrast, our participants rested completely after their HIFT bout. Without an active portion in our recovery period, it would be very difficult to make comparisons to the 200-m swim. Another important distinction is that High-intensity Interval Training and High-intensity Functional Training are two very different training modalities. HIIT incorporates high-intensity intervals (e.g., > 85% HRmax) separated by pre-defined rest intervals, whereas the intensity and rest intervals during HIFT are most often self-selected (i.e., effort and rest breaks occur at the participant’s discretion). Making comparisons to HIIT would not offer any greater insight than making comparisons to traditional resistance or aerobic training sessions. For this reason, we limited our comparisons to studies that specifically looked at HRV and catecholamine responses to HIFT. Further, the authors do not specify where data collection occurred; whether it was in a laboratory or the swimmers’ normal training facility. An important point in our discussion was that we observed differences between the laboratory setting and our athletes’ training facility in the HRV measures we collected. Thus, other than stating that Clemente-Suárez and Arroyo-Toledo also measured HRV in athletes surrounding exercise, it is not clear how their findings might be related or offer insight to the findings of the present investigation.
provide a conclusion that specifically responds to the objectives of the study.
We have revised the conclusion to be less speculative and focus on stating the main findings of our study and their relevance towards future research.
Round 2
Reviewer 2 Report
My first review asked the authors to revise the work, providing more persuasive arguments for doing the work. I though there was some good data but the work came across as a fishing exercise to find something useful. I offered the authors the chance to revise the work accordingly. The revised article has some useful additions but suffers from the same limitations. The argument that effect sizes are not worth reporting because they are too small is not strong. The decision to analyse by gender and make gender a leading message in the abstract remains a limitation.
As we have been through one round of revisions, I feel the paper has not moved sufficiently close to publication.